# Automated Parking Space Allocation during Transition with both Human-Operated and Autonomous Vehicles

**Mingkang Wu [1], Haobin Jiang [1],\* and Chin-An Tan [2]**

[1] School of Automotive and Traffic Engineering, Jiangsu University, Zhenjiang 212013, China; 2221804039@stmail.ujs.edu.cn

[2] Department of Mechanical Engineering, Wayne State University, 5050 Anthony Wayne Drive, Detroit, MI 48202, USA; tan@wayne.edu

\* Correspondence: jianghb@ujs.edu.cn; Tel.: +86-13805283226

**Abstract:** As fully automated valet parking systems are being developed, there is a transition period during which both human-operated vehicles (HVs) and autonomous vehicles (AVs) are present in the same parking infrastructure. This paper addresses the problem of allocation of a parking space to an AV without conflicting with the parking space chosen by the driver of a HV. A comprehensive assessment of the key factors that affect the preference and choice of a driver for a parking space is established by the fuzzy comprehensive method. The algorithm then generates a ranking order of the available parking spaces to first predict the driver's choice of parking space and then allocate a space for the AV. The Floyd algorithm of shortest distance is used to determine the route for the AV to reach its parking space. The proposed allocation and search algorithm is applied to the examples of a parking lot with three designed scenarios. It is shown that parking space can be reasonably allocated for AVs.

**Keywords:** automated parking system; fuzzy comprehension evaluation; Floyd algorithm; human-operated vehicle; autonomous vehicle





## 1. Introduction

According to the International Parking Institute (IPI), the number of vehicles on the road will reach 2.5 billion in 2050 [1]. With this projected increase in the volume of vehicles, parking has become an emerging issue that affects not only drivers looking for parking spaces, but also city governments in their planning, particularly in urban areas where land resources are limited and constrained. It has been reported that about 30% of traffic backup in a typical downtown area is caused by drivers searching for parking spaces [2]. The expected increase in the number of vehicles likely implies more new drivers and drivers who are unskilled in parking, thus leading to more road congestion and increased waste of valuable manpower-hours and resources. In recent years, with the continuous advancement and development in computer and control technologies, automated parking has become feasible and various strategies have been proposed to help alleviate the unskilled parking problem [3,4]. Advances in V2X technology have also led related researchers to develop a more robust system of automated parking, namely the Automated Valet Parking System [5].

Compared with the automated parking system, the concept of automated valet parking system is based on V2X communication technology, which enables self-driving vehicles to interact and collaborate with an intelligent parking infrastructure during the entire parking space search process, from the entrance to the self-parking space [6]. Existing intelligent parking administration systems can detect the status of the parking spaces in real time through camera recognition, infrared sensing, and other technologies [7,8]. According to SAE (Society of Automotive Engineers) classification for autonomous driving levels, the automated valet parking system is classified as L4 autonomous driving, that is, under

certain scenarios, an equipped vehicle can complete all the driving tasks autonomously without the participation of a driver. Although self-driving cars are gaining popularity, related research and testing are far from the level of completion required to bring automated valet parking to fruition. Even if L4 level self-driving cars were to enter the market in the short term, there is likely going to be a long transition period during which both human-operated vehicles (HVs), i.e., with drivers, and autonomous vehicles (AVs), i.e., self-driving or driverless, co-exist, and augmented search strategies need to be developed for the AVs. Self-driving vehicles execute well-defined algorithms based on sensor information, while drivers make cognitive decisions based on perception and surveying of the surrounding environment. The objective of this paper is to investigate parking space allocation and route selection (i.e., path planning to the allocated space) for self-driving vehicles during such a transition period in which significant interactions between the two types of vehicles are expected.

### 1.1. Motivation for Research

In this paper, as shown in Figure 1, we define an intelligent parking infrastructure (which refers to either a parking lot or a parking structure) as one that is equipped with a central command station that receives information from all sensors and interacts with autonomous vehicles in real time. In an automated valet parking system, the central command station is capable of determining the number of available parking spaces through geomagnetic sensors and obtaining the locations of these parking spaces through a pre-stored layout of the parking infrastructure. After an AV enters the parking infrastructure, the central command station utilizes information from multiple cameras to provide a road map to the allocated parking space, enabling the self-parking to be efficient. However, as development and implementation of automated parking systems move forward, there will be a transition period to fully automated valet parking unless the city government phases out HVs and mandates purchase of AVs, which is very unlikely, or designates parking spaces only for AVs. Thus, in an intelligent parking infrastructure, there will generally be both self-driving and human-driving vehicles. Self-driving vehicles can interact with the parking infrastructure in real time based on V2X communication. Human-driving vehicles are limited by non-intelligent devices that disengage any interaction with the central command station.

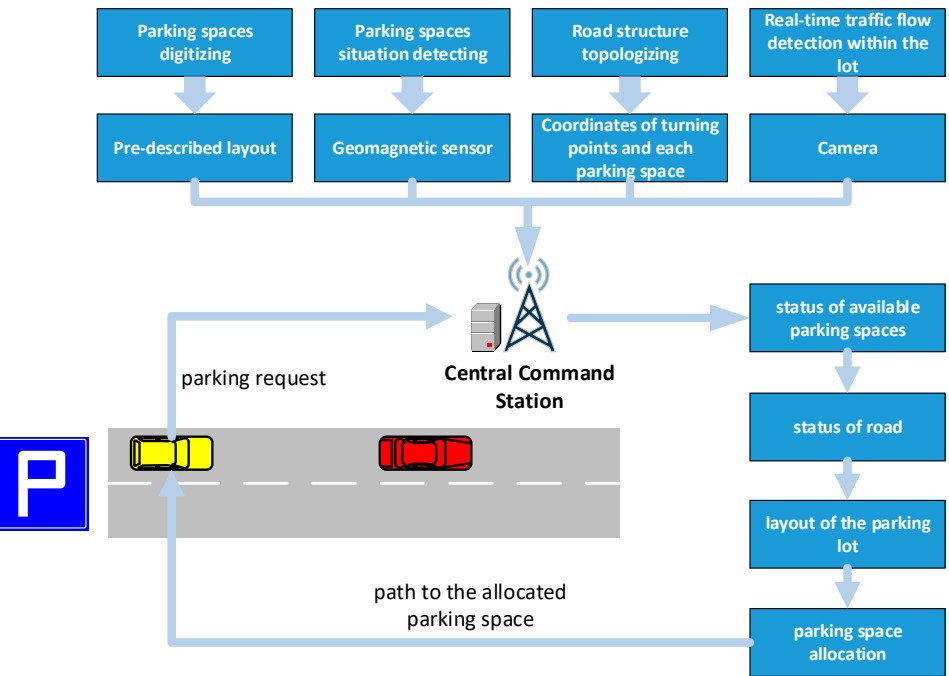

**Figure 1.** Flowchart of the entire process of parking space allocation and route selection for an autonomous vehicle.

Consider the scenario when both a HV and an AV are looking for parking spaces. In order for the central command station not to allocate the same parking space for the AV as the driver would choose, the station must be able to predict with high probability the choice of parking space that the driver would make. Liang et al. proposed four preference factors that could affect the choice of parking spaces for an individual driver [9]. Logit models, based on fuzzy logic theory, were established to describe driver preference selection, which were then used to improve parking experience and alleviate difficulties in parking [9,10]. Although fuzzy evaluation methods have been widely employed with proven success to determine factors, which are otherwise difficult to quantify, and make optimal choices incorporating expert opinions, there are only few published studies related to parking choice preference. In this paper, by adopting the fuzzy evaluation method, a preference ranking method of parking spaces in an intelligent parking infrastructure is established based on four representative factors that would affect the parking space selection of drivers.

In an automated valet parking system, route selection is also a key enabling technology [11,12]. Since the AV might not know a priori the overall layout of the parking infrastructure, it is necessary for the central command station to guide the self-driving vehicle to the allocated parking space according to the current conditions of the intended route. In general, in searching for a parking space, the shorter the path distance, the better. There are several methods to find the shortest path, such as the Dijkstra and Floyd algorithms [13,14]. Shi et al. [15] studied the shortest path planning problem of mobile robots based on the Floyd algorithm, focusing on the node selection problem for mobile robot path planning and determination of the weighting factor of each passable road. Experiments have also illustrated that the Floyd algorithm has the advantage of providing the shortest path selection for mobile robots [15]. Based on a known layout of the environment, Dijkstra algorithm can also efficiently find the shortest path between two points. Although the Floyd algorithm is slightly more time-consuming than the Dijkstra algorithm, it is a dynamic programming algorithm aimed to solve the shortest path problem between multiple source points. In a complex environment such as parking, the Floyd algorithm appears to be more suitable for our current problem of interest.

### 1.2. Organization and Contributions of This Paper

In this paper, state-of-the-art research of automated parking space allocation and route selection is reviewed in Section 2. Section 3 describes the four factors that affect the choice of parking space for drivers. In Section 4, a fuzzy comprehensive evaluation method is proposed to evaluate and score (i.e., rank) the available parking spaces in an intelligent parking infrastructure [16]. The fuzzy algorithm is based on first predicting which space the driver would select and then allocating one of the remaining spaces to the autonomous vehicle. In Section 5, the Floyd algorithm is introduced to provide path navigation for autonomous vehicles [14] according to the nodes of available parking spaces and road information in the parking infrastructure. Section 6 provides three examples of a parking lot on the campus of Jiangsu University to illustrate the step-by-step implementation of the proposed algorithm via Python. Concluding remarks of this work are given in Section 7.

The contributions of this paper are listed as follows:

1.　In this paper, a fuzzy comprehensive evaluation method is used to first predict the parking preference of a driver based on four main factors that influence their choice of parking spaces. This procedure prevents conflict of allocation of parking space to the AV with that chosen by the HV.
2.　Based on the node information in a parking infrastructure, the Floyd algorithm is used to provide path navigation for the AVs.
3.　The proposed methodology, combining fuzzy theory and the Floyd algorithm, provides a novel scheme for a central command station to assign parking spaces for AVs in the presence of HVs. The merit of this work thus provides a foundation for future work to investigate the problem of automated parking for multiple vehicles.

## 2. Review of Related Work

In 2007, the DARPA (Defense Advanced Research Projects Agency) Urban Challenge (DUC) initiated numerous research projects to address challenges of autonomous driving. Since then, a great deal of research has been conducted in related topics. In addition to highway and urban driving, automated valet parking is also of major interest to relieve drivers from the stress of parking [17].

In 2012, a parking information acquisition and release system was designed by Dou et al. based on the dynamic allocation algorithm of parking spaces in the parking lot [18]. The main idea is to optimize among the distribution of parking spaces arranged in a tabulated format and driving paths/distances. However, the algorithm can obtain the availability of parking spaces only after the human-operated vehicles have completed their parking. Thus, if there are more than two vehicles, at least one HV and one AV, choosing the same parking space and converging near that space, this scenario could lead to traffic congestion and the wasting of time and fuel consumption.

In 2012, Audi Corporation developed a parking guidance system that could assist and orchestrate the entire parking process [19]. The approach of the system includes three steps: scanning, positioning, and arranging parking [20]. The distance to the obstacle is detected by eight ultrasonic sensors installed in the bumper and the parking space is detected by 10 ultrasonic sensors. This parking system can only be executed if the speed of the vehicle is below 30 km/h. While this is an excellent demonstration of an automated valet parking system, during the transition period to fully autonomous driving, HVs will also be represent; the guidance system cannot allocate parking spaces to the HVs with no communication devices equipped and hence cannot complete the space allocation task for the AVs.

In 2016, Kotb et al. [21] proposed a system to reduce the time wasted in looking for parking spaces by offering guaranteed parking reservations with the lowest possible cost and searching time for drivers and the highest revenue and resource utilization for parking managers. However, they did not include other important factors that influence how drivers choose parking spaces, such as the safe locations of the parking spaces and the distance to reach the parking space.

In 2018, Tcheumadjeu et al. [22] presented an architecture for Automated Valet Parking (AVP) connected to cloud-based IoT services and mobile user interfaces. Autonomous vehicles can share information and data via the phones of their occupants. Moreover, under this communication architecture, some functions of the AVs, such as drop-off and pick-up, can be activated by phones. However, the paper only considers the interactions between users and AVs. In an actual scenario, many vehicles will be present at the same time, making efficient planning of traffic flow in the parking lot not possible by just integrating the shared information provided by the users.

In 2019, a new path planning system based on the Dijkstra algorithm was designed by Yu et al. [23]. The road occupancy factor is calculated and added to the path weight of the Dijkstra algorithm, and a shortest path is selected by comparing the weight of each path. However, the traffic situation in the parking lot is complex, and each section of the path may be occupied at any time. Hence, the optimal path planning in the parking lot requires a dynamic planning.

In 2020, a parking guidance system based on the multi-objective point A* algorithm was designed by Xiao et al. [24], which combines the distance factor between the entrance and the parking spaces and path planning using a heuristic function of the A* algorithm to quickly generate multiple driving paths and select the most efficient path. However, the heuristic function of the A* algorithm cannot take into consideration traffic congestion. If congestion happens, the time loss on queuing will reduce the efficiency of the system and subsequently increase the fuel consumption.

Therefore, in an intelligent parking infrastructure with both HVs and AVs involved, it is important to have a central command station to schedule the overall traffic flow and interface with autonomous vehicles. Moreover, it is necessary to consider in detail the fac-

tors that would affect parking space selection by the drivers and efficiently guide/allocate potentially available parking spaces for the AVs.

## 3. Factors for Choosing Optimal Parking Spaces

In an automated valet parking system, AVs are assumed to be able to interface with the central command station in an intelligent parking infrastructure. However, under the scenario in which there are both HVs and AVs in a parking infrastructure, no matter how intelligent the parking infrastructure is, the central command station cannot allocate parking spaces for the HVs which are not equipped with any communication devices. Based on this scenario, we consider the problem of a HV looking for a parking space in an intelligent parking infrastructure when an AV enters at the same time. The central command station, before allocating a parking space for the AV, must predict which parking space that the driver of the HV would choose and then allocate one of the remaining parking spaces to the autonomous vehicle. When a driver enters a parking infrastructure, his choice of parking space is often affected by his perception of various factors in the parking environment. Chen et al. [25] proposed six major factors that influence drivers to choose parking spaces: the walking distance from the parking spaces to the exit of the parking facility, the distance to the parking space from the entrance, the status of the path to the parking spaces, the status of the available parking spaces, parking safety, and shade under sunlight when parked outdoors. Since parking safety is somewhat implied in the status of the path for the AVs and the test case considered in this paper (see Section 6) is an outdoor parking lot with no shaded area, the last two factors will not be discussed in this paper. Thus, this paper focuses on the following four factors.

- *Walking distance*

Walking distance is the distance between the parking space and the exit of the parking infrastructure. For drivers, they likely prefer to arrive at their destination as quickly as possible after parking. Therefore, the shorter the walking distance, the more likely the parking space will be chosen.

- *Distance to parking*

Distance to parking is the distance between the entrance of the parking infrastructure and the parking space. For drivers and occupants, they generally prefer to arrive at their vehicles as quickly as possible, particularly when they forget something (e.g., phones, wallets) or need to place something back in the vehicle. Therefore, the shorter this distance, the more likely the parking space will be chosen.

- *Status of the path to the parking spaces*

Vehicles that break down (e.g., dead batteries) may block the path to some available spaces. Generally, lanes in parking infrastructures are narrow and often only one vehicle can pass through the lane. Lane occupancy increases traffic congestion and reduces traffic efficiency in the parking infrastructure. Thus, drivers are concerned with the status of the path leading to the available parking spaces and will likely choose those parking spaces that have clear paths.

- *Status of available parking spaces*

Ma [26] hypothesized that, in general, particularly new drivers are more inclined to choose available parking spaces with spaces on both sides unoccupied. The order of the priority of choice is followed by those spaces with one adjacent space unoccupied, those adjacent to the road, and finally those with spaces on both sides occupied.

Characteristics of parking spaces can be grouped into two main types: the cost type and the benefit type. For the four factors mentioned above, walking distance and distance to parking are cost types. Status of the path lane to parking spaces and status of available parking spaces are benefit types. The lower the value of the cost type factor, the higher the priority of the parking space would have. Contrarily, the higher the value of the benefit type factor, the higher priority of the parking space. Walking distance and distance to

parking can be described by measurable values based on the layout of the entire parking infrastructure. Fuzzy factors such as the status of lanes and available parking spaces can also be represented by quantitative values. Following [27], a value of three is assigned when the lane is occupied; nine when the lane to the parking space is clear; eight to available parking space whose both sides are unoccupied; seven to an available parking space with one adjacent space unoccupied; six to an available parking space adjacent to the road; five to an available parking space with spaces on both sides occupied. Note that these numbers are relative and their absolute values are irrelevant.

## 4. Optimal Parking Space Selection Model

For a driver, parking space selection is a decision based on his perception and assessment of the parking environment. As it is difficult to quantify these fuzzy concepts for humans, fuzzy theory is introduced to solve these types of problems. The concept of fuzzy theory was put forward by Professor Zadeh in 1965, aiming to quantify the uncertainty of issues [28]. Based on the fuzzy theory, the four factors affecting driver's choice of parking space selection, discussed in Section 3, are quantified. In this section, we will introduce how to assign and sort the weights of available parking spaces based on the Fuzzy Comprehensive Evaluation (FCE) method [28]. As a result, the order of parking space preference will be displayed with different weights.

*Procedure for Model Construction Based on FCE:*

(1) Establish a factor vector $U = (u_1, u_2, \ldots, u_m)$ where $u_i$ is the $i-$th factor that affects the parking selection. These factors usually have varying degrees of fuzziness. In this paper, the factor vector for evaluating parking preference is $U = (u_1, u_2, u_3, u_4)$, where $u_1$ represents "walking distance", $u_2$ "distance to parking", $u_3$ "status of lane to the parking spaces" and $u_4$ "status of available parking spaces".

(2) Establish an evaluation matrix $A$ whose $i$-th row, $(a_{i1}, a_{i2}, \ldots, a_{im})$, is the evaluation vector for the $i$-th available parking space, and $a_{ij}$ is the evaluation value for the $j-$th factor in $U$. In this paper, the evaluation vector for evaluating each factor in the vector $U$ is $(a_{i1}, a_{i2}, a_{i3}, a_{i4})$ for the $i$-th parking space, where $a_{i1}, a_{i2}$ are defined as the actual values of "walking distance" and "distance to parking" for evaluating $u_1$ and $u_2$, respectively; $a_{i3}$ and $a_{i4}$ are the scores under the evaluation standard discussed in Section 3 for evaluating the "status of lane to the parking spaces" $u_3$ and "status of available parking spaces" $u_4$, respectively.

(3) Establish the fuzzy comprehensive evaluation matrix $R = (r_{ij})_{n \times m}$ by normalizing $A$, where $n$ is the total number of targeted parking spaces that need to be evaluated and $m$ is the number of factors.

$$R = \begin{pmatrix} r_{11} & r_{12} & \cdots & r_{1m} \\ r_{21} & r_{21} & \cdots & r_{2m} \\ \vdots & \vdots & \ddots & \vdots \\ r_{n1} & r_{n2} & \cdots & r_{nm} \end{pmatrix} \tag{1}$$

The purpose of this normalization is to eliminate the impact caused by the differences in the orders of numbers of the physical measurements in the decision making process. For example, distance of "walking distance" maybe several meters while there is no particular unit for "status of parking spaces"; the orders of these numeric values are very different. The normalizations are:

$$r_{ij} = \frac{\min(a_{ij})}{a_{ij}}, \ i \in (1, 2, \ldots, n), \ j \in I_1 \tag{2}$$

$$r_{ij} = \frac{a_{ij}}{max(a_{ij})}, \ i \in (1, 2, \ldots, n), \ j \in I_2 \tag{3}$$

where, $I_1$ and $I_2$ represent the cost type factors and benefit factors, respectively [25].

(1) According to [29], to eliminate a potentially large number of combinatorial comparisons due to a large dimension of the factor vector, it is necessary to evaluate the independent relationship between every two factors by the pair-wise comparison matrix $B = (b_{ij})_{n \times m}$ which is defined by drivers with different driving experiences, where $b_{ii} = 0.5$, $b_{ij} + b_{ji} = 1$, and $b_{ij} \geq 0$. $b_{ij}$ represent the value of factor $i$ compared to the preference to factor $j$, $b_{ii}$ represents the value of a factor compared to the preference to itself. In this paper, we have collected opinions from twenty drivers to form the pair-wise comparison matrix. Our sample included five males and five females, with ages varying from 20 to 55 years old and with driving experiences ranging from less than one month to more than 35 years. This set of samples represents sufficiently broad variability.

(2) Define the weight vector of the factors: $w = (w_1, w_2, \ldots, w_n)$, where $w_i$ can be calculated by the least variance method (LVM) for further ranking priority of the factors based on the pair-wise comparison matrix. [30]:

$$w_i = \frac{1}{n} \left( \sum_{j=1}^{n} b_{ij} + 1 - \frac{n}{2} \right), \ i = (1, 2, \ldots, n) \tag{4}$$

(3) According to Equations (1)–(3) we established a priority vector $z_{i(w)}$, which can be described as follows:

$$z_{i(w)} = WAA_w(r_{i1}, r_{i2}, \ldots, r_{in}) = \sum_{j=1}^{n} R_{n \times m} w_i \tag{5}$$

where, $WAA_w(r_{i1}, r_{i2}, \ldots, r_{in})$ is the weighted arithmetic average operator. The higher the weight value of a certain parking space in $z_{i(w)}$, the more likely it would be selected by the driver [31].

## 5. Optimal Route Selection

Shortest path planning is very important in path navigation. Traffic efficiency can be improved by planning the shortest path based on the distance information between locations. Floyd algorithm is a classic dynamic programming algorithm that uses dynamic programming to find the shortest path between multiple source points in a given weighted graph. The algorithm aims to find the shortest path from one point to another [32].

Assume that the shortest path from a node $i$ to another node $j$ has no more than two possibilities, one is directly from $i$ to $j$, and the other is from $i$ through node $k$ to $j$. Hence, the dynamic transfer function of this algorithm is: $dis(i,j) = min\left(D_{i,j}, D_{i,k} + D_{k,j}\right)$, where $dis(i,j)$ represents the shortest distance between node $i$ and node $j$. The specifics of our proposed algorithm is shown as follows:

(1) Number each parking location in the map as a node;

(2) Initialize an adjacent matrix $D_{p \times q}$, where $D_{i,j}$ represents the distance between node $i$ and $j$; if $i$ and $j$ are not adjacent, $D_{i,j}$ will be assigned with $\infty$; if $i = j$, the value of $D_{i,j}$ will be 0.

$$D_{p \times q} = \begin{pmatrix} 0 & D_{1,2} & D_{1,3} & \cdots & D_{1,\,q-1} & D_{1,q} \\ D_{2,1} & 0 & D_{2,3} & \cdots & D_{2,q-1} & D_{2,q} \\ D_{3,1} & D_{3,2} & 0 & \cdots & D_{3,q-1} & D_{3,q} \\ \vdots & \vdots & \vdots & \ddots & \vdots & \vdots \\ D_{p-1,1} & D_{p-1,2} & D_{p-1,3} & \cdots & 0 & D_{p-1,q} \\ D_{p,1} & D_{p,2} & D_{p,3} & \cdots & D_{p,q-1} & 0 \end{pmatrix} \tag{6}$$

(3) Update the transfer function. For example, consider three nodes $i$, $j$, and $k$ in the map, where $k$ is an intermediate node of $i$ and $j$, and $i$ and $j$ represents the start

point and the end point, respectively. The transfer function $dis(i,j)$ is updated by $dis(i,j) = \min\left(D_{i,j},\ D_{i,k} + D_{k,j}\right)$ to select the smaller of the two distances from $i$ to $j$ and from $i$ to $k$ to $j$. The Floyd algorithm finds an intermediate node $k$ to determine whether there is a shorter distance through this node $k$.

## 6. Example: Results and Discussion

The example we consider is a single-entrance parking lot, see Figure 2. The layout of the parking lot has been redesigned according to the Engineering Construction Industry Standard JGJ100-98 and the layout of the example parking lot is shown in Figure 3. Since the entrance is located in the lower right of the entire layout, we define the coordinate system with the origin at the lower right corner, as shown in Figure 4.

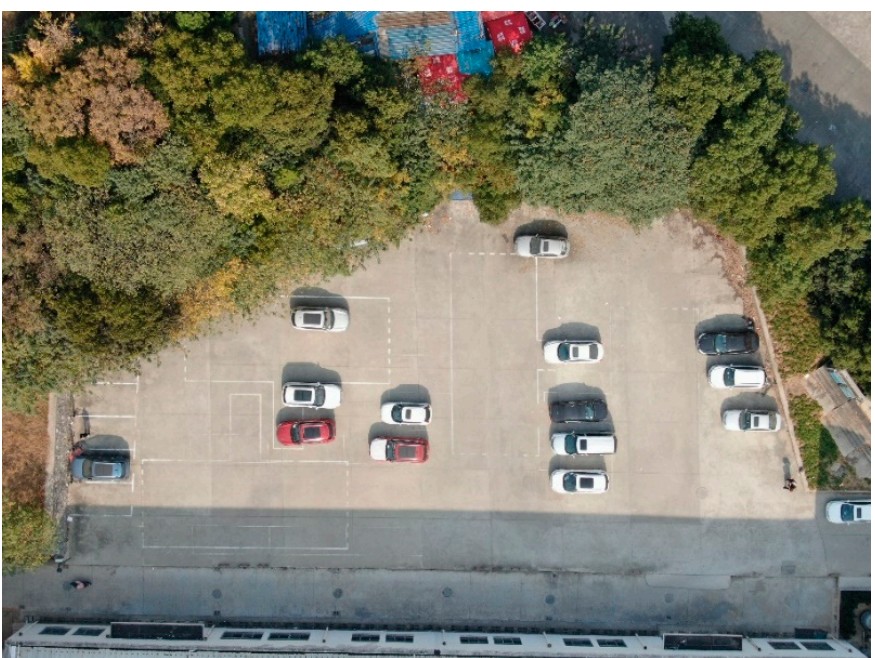

**Figure 2.** Aerial view of the parking lot of our example problem.

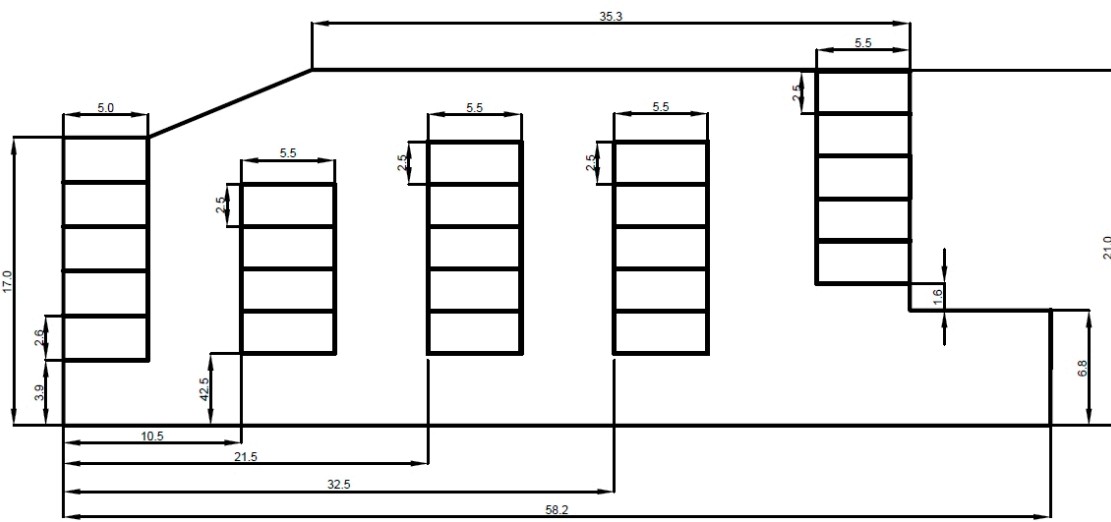

**Figure 3.** Layout of the parking lot (distance unit in meter).

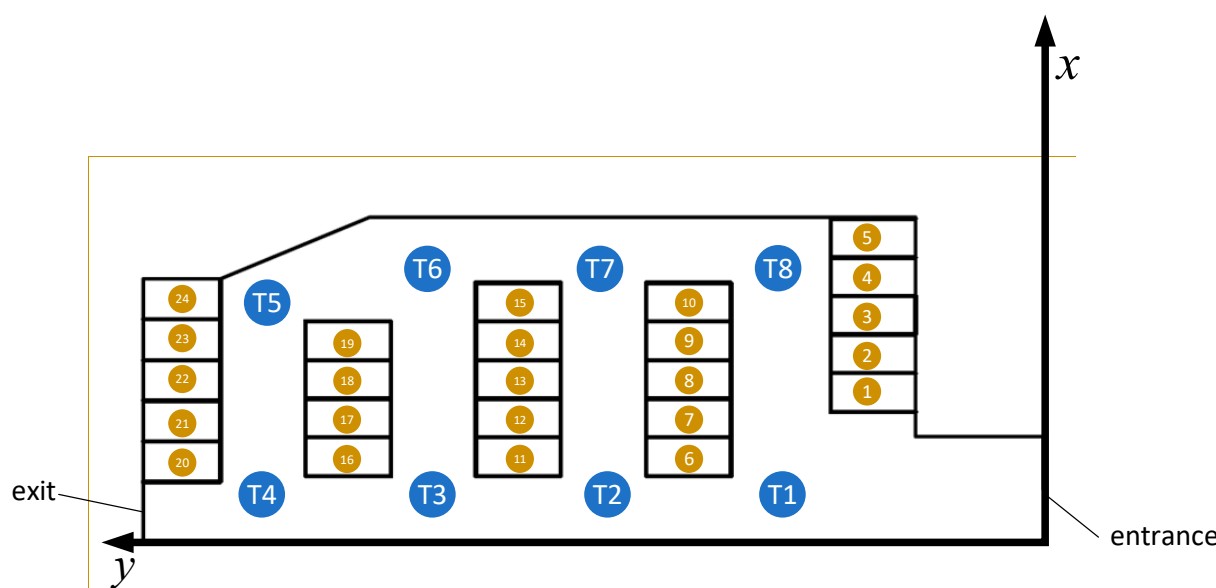

**Figure 4.** Layout of nodes in the parking lot and reference to a coordinate system.

As shown in Figure 4, in order to design the route selection, we denote each parking space and each turning intersection as a node (1–24) and a number (T1–T8). Additionally, considering the differences in sizes of vehicles entering the parking lot, we define the coordinates of each of the nodes and intersection numbers as the centered position of the parking space and the central line of the route at the turning. The coordinates of all nodes are listed in Table 1.

**Table 1.** Coordinates in metric units of all nodes in test graph

| T1 | (2.1, 17.0) | 1 | (9.7, 17.0) | 9 | (13.0, 17.0) | 17 | (8.0, 39.5) |
|----|-------------|---|-------------|---|--------------|----|-------------|
| T2 | (2.1, 28.5) | 2 | (12.2, 17.0) | 10 | (15.5, 17.0) | 18 | (10.5, 39.5) |
| T3 | (2.1, 39.5) | 3 | (14.7, 17.0) | 11 | (5.5, 28.5) | 19 | (13.0, 39.5) |
| T4 | (2.1, 50.4) | 4 | (17.2, 17.0) | 12 | (8.0, 28.5) | 20 | (5.2, 50.5) |
| T5 | (15.7, 50.4) | 5 | (19.7, 17.0) | 13 | (10.5, 28.5) | 21 | (7.8, 50.5) |
| T6 | (19.7, 39.5) | 6 | (5.5, 17.0) | 14 | (13.0, 28.5) | 22 | (10.4, 50.5) |
| T7 | (19.7, 28.5) | 7 | (8.0, 17.0) | 15 | (15.5, 28.5) | 23 | (13.1, 50.5) |
| T8 | (19.7, 17.0) | 8 | (10.5, 17.0) | 16 | (5.5, 39.5) | 24 | (15.7, 50.5) |

In this section, we have designed three specific parking scenarios to test our proposed algorithm via Python. In the following, we will take scenario 1 as an example to explain the application of our method in detail. As shown in Figure 5, red nodes represent the parking spaces that are occupied, brown ones are those that are currently available, and the light blue node (node 11) represents the parking space that is being parked by a vehicle. A blue vehicle, which is an AV, is just entering into the intelligent parking lot and sending a parking request to the central command station. Meanwhile, an orange vehicle, which is a HV and cannot communicate with the central command station, is seeking an available parking space. Under this situation, the problem is to allocate an available parking space for the blue autonomous vehicle by the central command station.

In order to solve this problem, we need to first determine which parking space that the driver of the orange vehicle would choose and offer the shortest path between the entrance and the allocated parking space. The overview scheme is described as a flowchart (Figure 6). The specific steps in the parking spaces selection and the shortest path planning are shown as follows:

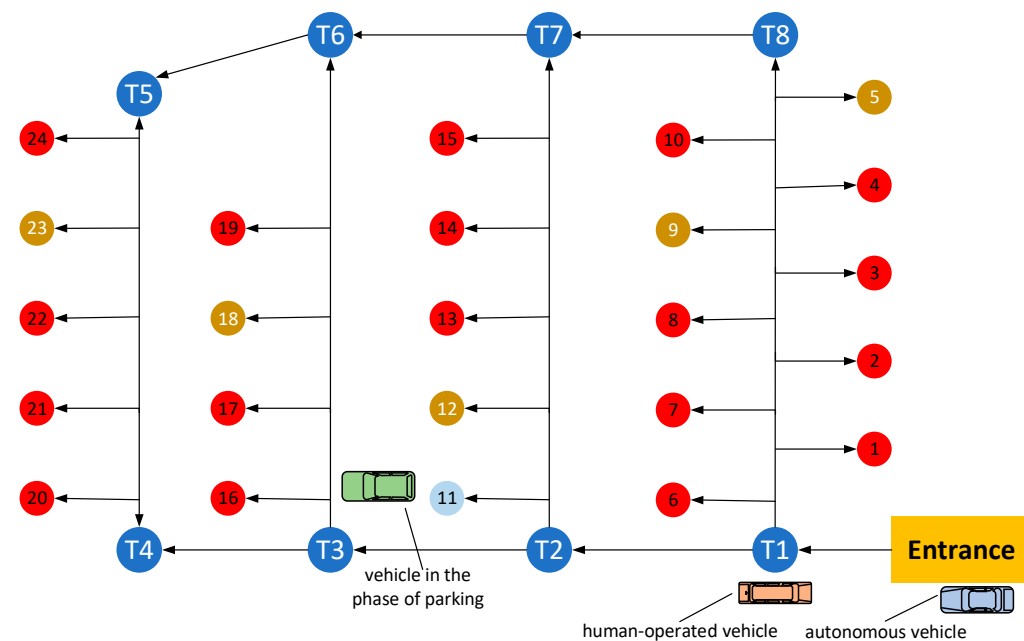

**Figure 5.** Example of a parking scenario 1.

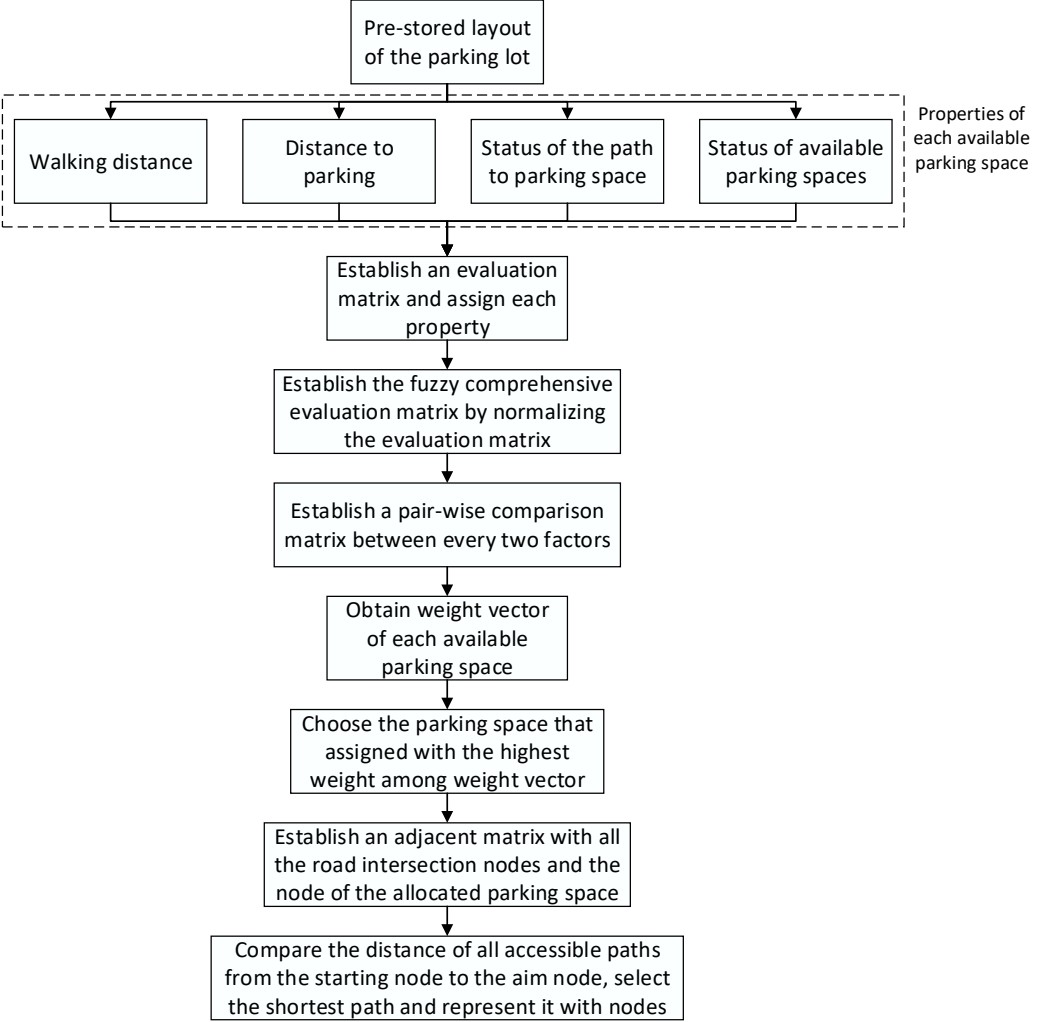

**Figure 6.** Flowchart for parking space selection and the shortest path planning.

Step 1: Set the properties table of all available parking spaces currently in the example parking lot. As shown in Table 2.

**Table 2.** Properties of the available parking spaces in the parking lot of scenario 1.

| Parking Space | Walking Distance (m) | Distance to Parking (m) | Status of Parking | Status of Available Parking Spaces |
|---|---|---|---|---|
| 5 | 51.0 | 36.7 | clear | adjacent to the road |
| 9 | 44.3 | 30.0 | clear | both sides are occupied |
| 12 | 27.8 | 36.5 | occupied | both sides are occupied |
| 18 | 41.8 | 27.5 | clear | both sides are occupied |
| 23 | 11.1 | 63.6 | clear | both sides are occupied |

For the walking distance and distance to parking, distances in metric units are used in evaluating the values in matrix $A$, as defined in Section 4. For the status of lane and status of available parking spaces, fuzzy evaluation is assigned with values proposed in Section 3. Based on the values from Table 2, the evaluation matrix $A$ is:

$$A = \begin{pmatrix} 51.0 & 36.7 & 9.0 & 6.0 \\ 44.3 & 30.0 & 9.0 & 5.0 \\ 27.8 & 36.5 & 3.0 & 5.0 \\ 41.8 & 27.5 & 9.0 & 5.0 \\ 11.1 & 63.6 & 9.0 & 5.0 \end{pmatrix}$$

Step 2: Normalize the evaluation matrix $A$ to form the fuzzy comprehensive evaluation matrix $R = (r_{ij})_{n \times m}$ according to Equations (2) and (3):

$$R = \begin{pmatrix} 0.217 & 0.749 & 1.000 & 1.000 \\ 0.250 & 0.917 & 1.000 & 0.833 \\ 0.399 & 0.753 & 0.333 & 0.833 \\ 0.266 & 1.000 & 1.000 & 0.833 \\ 1.000 & 0.432 & 1.000 & 0.833 \end{pmatrix}$$

Step 3: Establish the pair-wise comparison matrix by the comparison method, based on the opinions of our sample set of twenty drivers with a range of driving experiences, as described in Section 4. The following shows the response from one of the drivers:

$$B_i = \begin{pmatrix} 0.5 & 0.8 & 0.9 & 0.4 \\ 0.2 & 0.5 & 0.1 & 0.4 \\ 0.1 & 0.9 & 0.5 & 0.9 \\ 0.6 & 0.6 & 0.1 & 0.5 \end{pmatrix}$$

where, $i = 1, \ldots, 10$ for our data set. As shown in $B_i$, all diagonal values are 0.5 as each factor complements itself. The values of $b_{12} = 0.8$ and $b_{21} = 0.2$ indicate that in the view of this driver, the factor "walking distance" is more important than "distance to parking". "distance to parking" is still a factor that the driver would consider when choosing a parking space, but it is just less important.

Step 4: Obtain the weight vector for the pair-wise comparison matrix in Step 3 according to Equation (4):

$$w_i = (0.40, \ 0.05, \ 0.35, \ 0.20)$$

Applying Equation (4), the pair-wise comparison matrix from the twenty drivers is converted into the following weight matrix:

$$
w = \begin{pmatrix}
0.400 & 0.050 & 0.350 & 0.200 \\
0.200 & 0.325 & 0.450 & 0.275 \\
0.500 & 0.125 & 0.175 & 0.175 \\
0.175 & -0.050 & 0.625 & 0.750 \\
0.150 & 0.375 & 0.250 & 0.225 \\
0.225 & 0.125 & 0.250 & 0.400 \\
0.125 & 0.550 & 0.350 & 0.350 \\
0.375 & 0.100 & 0.350 & 0.175 \\
0.125 & 0.225 & 0.375 & 0.250 \\
0.225 & 0.200 & 0.400 & 0.225 \\
0.250 & 0.075 & 0.475 & 0.200 \\
0.425 & 0.075 & 0.475 & 0.200 \\
0.425 & 0.075 & 0.325 & 0.175 \\
0.125 & 0.125 & 0.425 & 0.325 \\
0.050 & 0.275 & 0.225 & 0.450 \\
0.175 & 0.250 & 0.275 & 0.300 \\
0.300 & 0.125 & 0.250 & 0.325 \\
0.325 & 0.175 & 0.100 & 0.425 \\
0.300 & 0.000 & 0.450 & 0.250 \\
0.150 & 0.150 & 0.425 & 0.325 \\
0.100 & 0.300 & 0.325 & 0.275
\end{pmatrix}
$$

To make the above matrix representative of the preference of most drivers, we remove the largest and smallest values of each factor in the above matrix and average the remaining values to obtain an average weight vector:

$$\widetilde{w} = (0.233,\ 0.170,\ 0.336,\ 0.286)$$

Step 5: With the average weight vector and Equation (5), the priority vector that contains the weights of every available parking space in the given situation can be obtained:

$$z_i(w) = (0.799,\ 0.788,\ 0.571,\ 0.806,\ 0.881)$$

Step 6: Ranking the priorities of the available parking spaces in this example parking lot according to $z_i(w)$ gives: $z_{23}(w) > z_{18}(w) > z_5(w) > z_9(w) > z_{12}(w)$.

Based on step 6, parking space 23 has the highest priority for the driver of the orange vehicle. Thus, parking space 18 will be allocated to the blue autonomous vehicle by the central command station. By analyzing the preferences of the human driver, the algorithm is able to provide the optimal parking space for the autonomous vehicle. Since an intelligent parking lot system cannot influence the choice of the driver, a suboptimal parking space, which is space 18, is allocated to autonomous vehicle in this situation.

Step 7: Establish the adjacent matrix with all the road intersection nodes and the node of parking space 18 for route selection based on the adjacent matrix assignment rule according to Equation (6) in Section 5:

$$
D = \begin{pmatrix}
0 & 11.5 & \infty & \infty & \infty & \infty & \infty & 17.6 & \infty \\
\infty & 0 & 11.0 & \infty & \infty & \infty & 17.6 & \infty & \infty \\
\infty & \infty & 0 & 10.9 & \infty & 17.6 & \infty & \infty & 8.4 \\
\infty & \infty & \infty & 0 & 13.6 & \infty & \infty & \infty & \infty \\
\infty & \infty & \infty & 13.6 & 0 & \infty & \infty & \infty & \infty \\
\infty & \infty & \infty & \infty & 11.6 & 0 & \infty & \infty & 9.2 \\
\infty & \infty & \infty & \infty & \infty & 11.0 & 0 & \infty & \infty \\
\infty & \infty & \infty & \infty & \infty & \infty & 11.5 & 0 & \infty \\
\infty & \infty & \infty & \infty & \infty & \infty & \infty & 0 & 0
\end{pmatrix}
$$

As shown in matrix $D$, there is only one passable path from T1 to parking space 18: starting from T1, go through T2, subsequently arrive at T3, then turn right to parking space 18, which is also the shortest path, with a total distance of 30.9 m. Through this method, under the scenario of the presence of one HV and one AV in this example, we simultaneously predict the parking space preference of the driver and allocate a parking space to the AV through V2X communication based on the driver preferences. Moreover, a parking lot may be constrained with traffic rules (e.g., one way road) and limited space, the adjacent matrix $D$ can, in addition to providing the shortest route navigation, also reflect the constraints of the traffic rules and other conditions.

Using the same approach as used in scenario one, we can also solve the similar issues, such as in the following two scenarios:

Scenario 2:

Step 1: Set the properties table of all available parking spaces currently in the example parking scenario (Figure 7). As shown in Table 3.

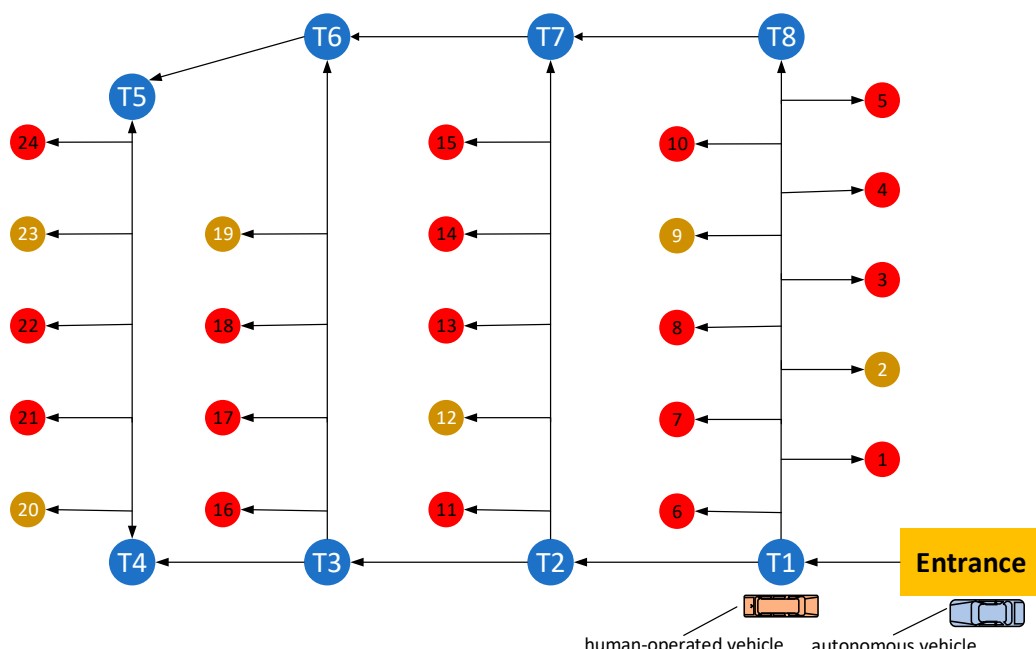

**Figure 7.** Example of a parking scenario 2.

**Table 3.** Properties of the available parking spaces in the parking lot of scenario two.

| Parking Space | Walking Distance (m) | Distance to Parking (m) | Status of Parking | Status of Available Parking Spaces |
|---|---|---|---|---|
| 2 | 43.5 | 29.2 | clear | both sides are occupied |
| 9 | 44.3 | 30.0 | clear | both sides are occupied |
| 12 | 27.8 | 36.5 | clear | both sides are occupied |
| 19 | 21.8 | 42.5 | clear | adjacent to the road |
| 20 | 3.2 | 55.7 | clear | adjacent to the road |
| 23 | 11.1 | 63.6 | clear | both sides are occupied |

Based on the values from Table 3 and assignment rule in Section 3, the evaluation matrix $A$ is:

$$A = \begin{pmatrix} 43.5 & 29.2 & 9.0 & 5.0 \\ 44.3 & 30.0 & 9.0 & 5.0 \\ 27.8 & 36.5 & 9.0 & 5.0 \\ 21.8 & 42.5 & 9.0 & 6.0 \\ 3.2 & 55.7 & 9.0 & 6.0 \\ 11.1 & 63.6 & 9.0 & 5.0 \end{pmatrix}$$

Step 2: Normalize the evaluation matrix $A$ to form the fuzzy comprehensive evaluation matrix $R = (r_{ij})_{n \times m}$ according to Equations (2) and (3):

$$R = \begin{pmatrix} 0.074 & 1.000 & 1.000 & 0.833 \\ 0.072 & 0.973 & 1.000 & 0.833 \\ 0.115 & 0.799 & 1.000 & 0.833 \\ 0.147 & 0.687 & 1.000 & 1.000 \\ 1.000 & 0.524 & 1.000 & 1.000 \\ 0.288 & 0.459 & 1.000 & 0.833 \end{pmatrix}$$

Step 3: With the average weight vector calculated in scenario one and Equation (5), the priority vector that contains the weights of every available parking space in scenario two can be obtained:

$$z_i(w) = (0.761, 0.756, 0.737, 0.773, 0.944, 0.719)$$

Step 4: Ranking the priorities of the available parking spaces in this example parking lot according to $z_i(w)$ gives: $z_{20}(w) > z_{19}(w) > z_2(w) > z_9(w) > z_{12}(w) > z_{23}(w)$.

Based on Step 4, parking space 20 has the highest priority for the driver of the orange vehicle based on scenario two. Thus, parking space 19 will be allocated to the blue autonomous vehicle by the central command station.

Scenario 3:

Step 1: Set the properties table of all available parking spaces currently in the example parking scenario (Figure 8). As shown in Table 4.

Based on the values from Table 4 and assignment rule in Section 3, the evaluation matrix $A$ is:

$$A = \begin{pmatrix} 51.0 & 36.7 & 9.0 & 6.0 \\ 41.8 & 27.5 & 9.0 & 5.0 \\ 25.3 & 34.0 & 9.0 & 6.0 \\ 27.8 & 36.5 & 9.0 & 7.0 \\ 21.8 & 42.5 & 9.0 & 6.0 \\ 3.2 & 55.7 & 9.0 & 6.0 \\ 11.1 & 63.6 & 9.0 & 5.0 \end{pmatrix}$$

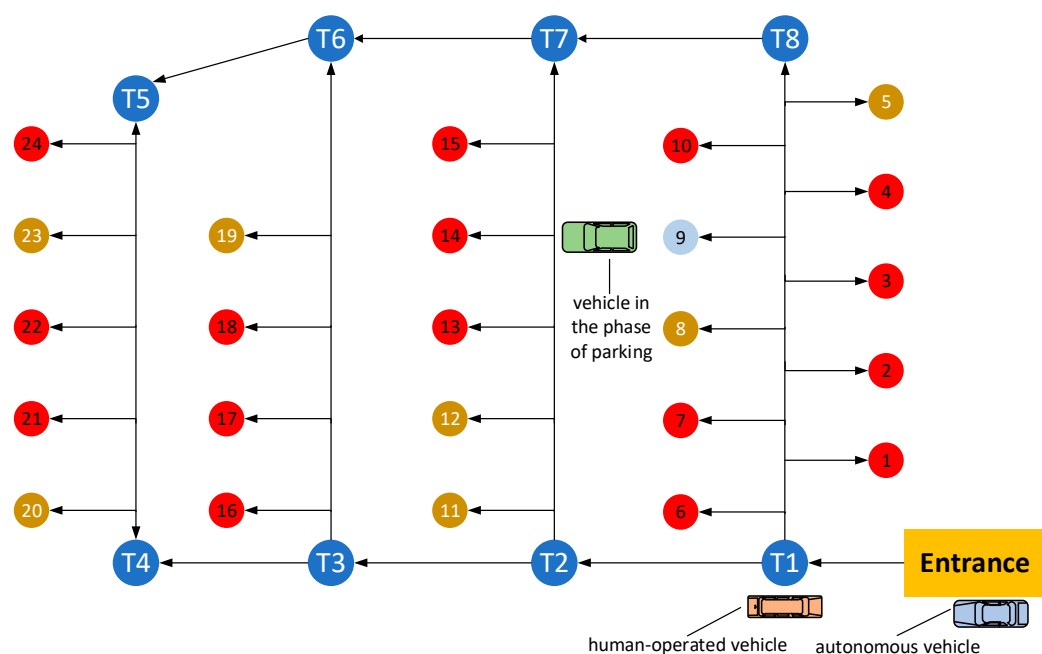

**Figure 8.** Example of a parking scenario three.

**Table 4.** Properties of the available parking spaces in the parking lot of scenario two.

| Parking Space | Walking Distance (m) | Distance to Parking (m) | Status of Parking | Status of Available Parking Spaces |
|---|---|---|---|---|
| 5 | 51.0 | 36.7 | clear | adjacent to the road |
| 8 | 41.8 | 27.5 | clear | both sides are occupied |
| 11 | 25.3 | 34.0 | clear | adjacent to the road |
| 12 | 27.8 | 36.5 | clear | one adjacent space unoccupied |
| 19 | 21.8 | 42.5 | clear | adjacent to the road |
| 20 | 3.2 | 55.7 | clear | adjacent to the road |
| 23 | 11.1 | 63.6 | clear | both sides are occupied |

Step 2: Normalize the evaluation matrix $A$ to form the fuzzy comprehensive evaluation matrix $R = (r_{ij})_{n \times m}$ according to Equations (2) and (3):

$$R = \begin{pmatrix} 0.063 & 0.749 & 1.000 & 0.857 \\ 0.077 & 1.000 & 1.000 & 0.714 \\ 0.126 & 0.809 & 1.000 & 0.857 \\ 0.115 & 0.753 & 1.000 & 1.000 \\ 0.147 & 0.647 & 1.000 & 0.857 \\ 1.000 & 0.494 & 1.000 & 0.857 \\ 0.288 & 0.432 & 1.000 & 0.714 \end{pmatrix}$$

Step 3: With the average weight vector calculated in scenario one and Equation (5), the priority vector that contains the weights of every available parking space in scenario three can be obtained:

$$z_i(w) = (0.723, 0.728, 0.748, 0.777, 0.725, 0.898, 0.681)$$

Step 4: Ranking the priorities of the available parking spaces in this example parking lot according to $z_i(w)$ gives: $z_{20}(w) > z_{12}(w) > z_{11}(w) > z_8(w) > z_{19}(w) > z_5(w) > z_{23}(w)$.

Based on step 4, parking space 20 has the highest priority for the driver of the orange vehicle based on scenario two. Thus, parking space 12 will be allocated to the blue autonomous vehicle by the central command station.

As can be seen from the above three test examples, people are more inclined to choose the available parking space near the exit of the parking lot.

In this paper, to validate of the proposed algorithm, we have collected opinions from fifty drivers on their most preferred parking spaces to check against the data from the twenty drivers mentioned in Step 3. Figure 9 shows that, in addition to the preferred choice of space 23, 20 and 20 in designed scenarios one, two and three, respectively, there are only four, six and four other choices among the fifty drivers who are surveyed. The accuracy of the prediction of the proposed algorithm is thus, 92%, 88% and 92%, respec-tively, in the three test cases.

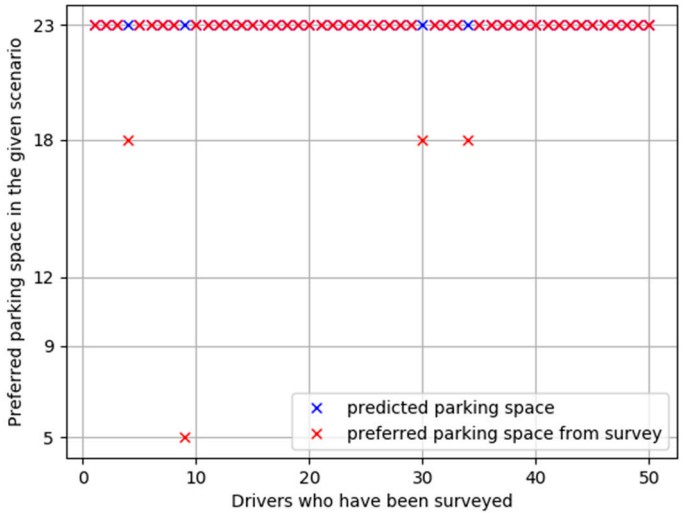

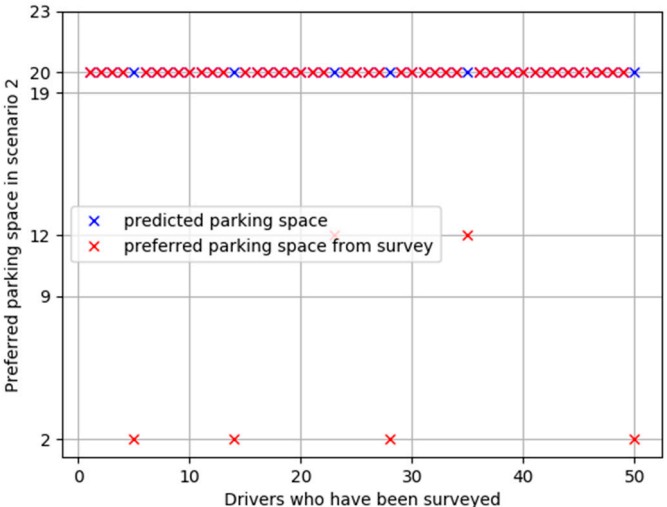

**Figure 9.** *Cont.*

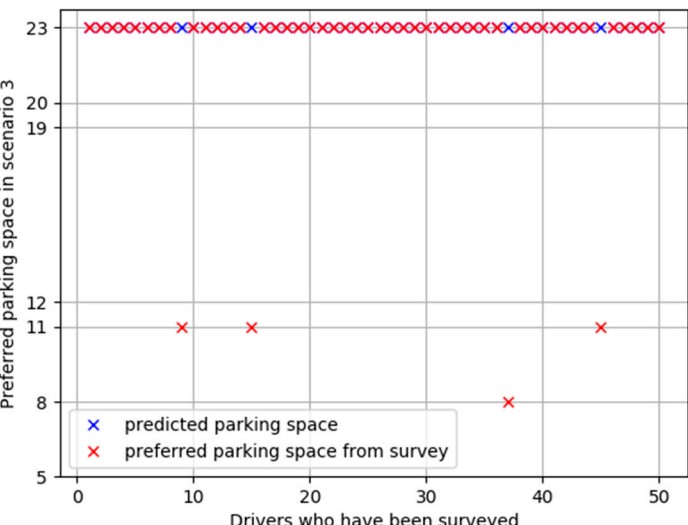

**Figure 9.** Comparison between the predicted parking space preference and the opinions of the most preferred parking space from fifty drivers in three example scenarios.

## 7. Conclusions and Future Work

In this paper, we studied the problem of automated parking space allocation during the transition period when both human-operated vehicles (HVs) and autonomous (driverless) vehicles (AVs) are present simultaneously in an intelligent parking infrastructure. Since not all vehicles will be autonomous in a near foreseeable future and fully automated parking systems are still being developed, the problem of interest, thus, has important and relevant applications. Based on four key factors that affect the choice of parking spaces for drivers, the fuzzy comprehensive evaluation (FCE) method is used to first predict the driver's choice of parking space, since a HV cannot communicate with the central command station. This prediction then allows an intelligent parking system to allocate the remaining available parking spaces to the AVs without conflict arising with the HVs. Furthermore, by incorporating information of the parking spaces and road map of the parking facility, an adjacent matrix in the Floyd algorithm is then developed to determine the shortest feasible paths to arrive at the parking spaces safely. The proposed algorithms are applied to three parking lot scenarios in which both a HV and an AV are entering. It is shown that a parking space can be reasonably allocated to the AV after a prediction of the driver's choice of parking space.

This work lays the foundation for future investigation into the automated parking problem with multiple HVs and AVs. Moreover, since the test parking lot discussed in this paper is not very large, it is likely that drivers can observe the entire layout and the situation from the entry point. More complexities of the parking infrastructure, such as multiple entrances, exits, and levels, can be considered in the future. A faster and more robust dynamic planning method needs to be developed.

**Author Contributions:** M.W. and H.J. designed the algorithm; C.-A.T. verified the rationality of the algorithm; M.W. performed the software simulation; H.J. and C.-A.T. analyzed the results of the simulation; M.W. wrote the paper with help of H.J. and C.-A.T.; C.-A.T. completed the final editing. All authors have read and agreed to the published version of the manuscript.

**Funding:** This work is financially supported by The Major University Nature Science Research Project of Jiangsu Province (No.16KJA580001).

**Institutional Review Board Statement:** Not applicable.

**Informed Consent Statement:** Not applicable.

**Data Availability Statement:** Not applicable.

**Conflicts of Interest:** The authors declare no conflict of interest.

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
