# Peer review of "Automated Parking Space Allocation during Transition with both Human-Operated and Autonomous Vehicles"

_applsci, doi:10.3390/app11020855_

Round 1

Reviewer 1 Report

The autonomous (driverless) vehicles (AVs) are not yet a reality. The question is: Why in such a future the AVs will not have their specific places? Such a case may conduct the AVs by sensors in one of them. It is a practical question...which in my opinion must be answered...

The choice between the two algorithms is not clear.

Reviewer 2 Report

The manuscript introduces a method for navigation of autonomous vehicles to a parking place. The method was designed for parking lots that can be used by both autonomous cars and human drivers. This topic is of current research interest, and the work is well motivated.

Unfortunately, the effectiveness and robustness of the proposed method were not sufficiently verified by experiments. The Authors have just presented one example of calculations for a specific parking scenario with three vehicles. Before publication, the method has to be confirmed in extensive experiments. Moreover, the method should be compared with other approaches from the literature.

The Authors have collected opinions from only ten drivers to form the pair-wise comparison matrix. The considered sample of drivers is too small.

Also, the scalability of the method has to be discussed. The centralized approach is an essential drawback of the method in the case of parking applications in larger areas.

The method seems to give a strong preference for vehicles driven by humans. This issue needs additional discussion.

In this paper, we have collected opinions from ten drivers to form the pair-wise comparison matrix

Line 256: It is necessary to explain the “cost type factors better and benefit factors” with more details.

Line 302: The dimensions of matrix D (m x n) are incorrectly defined. The authors have stated that ??,? represents the distance between node ? and j. As explained earlier in the manuscript, “? is the number of factors.” Thus, the definitions are inconsistent.

Figure 3: The accuracy of dimensions is too high. It is not necessary to present the distances in millimeters.

Figure 4: The meaning of T1 – T8 should be explained in the text.

Reviewer 3 Report

The issues presented in the article are very interesting, but after reading the text carefully, the following issues appeared:

- lines 75-86: "In this paper, adopting the fuzzy evaluation method, preference ranking of parking spaces in an intelligent parking infrastructure is established based on four representative factors that would affect the parking space selection of drivers." - the presented approach is a questionable issue, because in the case of a small number of vacancies, drivers' preferences are irrelevant, as it is important to find any free space.

- lines 247-251: how are the "ai3" and "ai4" values calculated? There is only a general description in section 3, which would need to be covered in more detail with an indication of the rules.

- lines 258-261: "For example, mass of a vehicle may be several thousand kg while velocity may be few m/s;" - why mass of vehicle and velocity were given as examples if these parameters are not included in the presented calculations?

- lines 266-270: description needs to be expanded.

- formuła w linii 302: whether the variables "m" and "n" are the same as on lines 253-25?

- formulas (1) and (2): which formula was used in the presented method?

- some formulas are missing numbering, e.g. in lines 256, 302.

- fig.4: the description of "T1", "T2", ... marks is missing.

- line 514: position 33 is empty;

General remarks:

- Section Methodology should be described in more detail.
- The results should be described in more detail in Section Example.

Round 2

Reviewer 2 Report

The authors have improved the paper by considering all suggestions satisfactory. I believe the paper can be accepted for publication.

Reviewer 3 Report

The introduced changes and additions are satisfactory.